# Loneliness among people with severe mental illness during the COVID-19 pandemic: Results from a linked UK population cohort study

Paul Heron[1]*, Panagiotis Spanakis[1], Suzanne Crosland[1], Gordon Johnston[2], Elizabeth Newbronner[1], Ruth Wadman[1], Lauren Walker[1], Simon Gilbody[1,3], Emily Peckham[1]

**1** Mental Health and Addiction Research Group, University of York, York, United Kingdom, **2** Independent Peer Researcher, United Kingdom, **3** Hull York Medical School, York, United Kingdom

* paul.heron@york.ac.uk

## Abstract

### Aim/Goal/Purpose

Population surveys underrepresent people with severe mental ill health. This paper aims to use multiple regression analyses to explore perceived social support, loneliness and factor associations from self-report survey data collected during the Covid-19 pandemic in a sample of individuals with severe mental ill health.

### Design/Methodology/Approach

We sampled an already existing cohort of people with severe mental ill health. Researchers contacted participants by phone or by post to invite them to take part in a survey about how the pandemic restrictions had impacted health, Covid-19 experiences, perceived social support, employment and loneliness. Loneliness was measured by the three item UCLA loneliness scale.

### Findings

In the pandemic sub-cohort, 367 adults with a severe mental ill health diagnosis completed a remote survey. 29–34% of participants reported being lonely. Loneliness was associated with being younger in age (adjusted $OR$ = -.98, $p$ = .02), living alone (adjusted $OR$ = 2.04, $p$ = .01), high levels of social and economic deprivation (adjusted $OR$ = 2.49, $p$ = .04), and lower perceived social support ($B$ = -5.86, $p$ < .001). Living alone was associated with lower perceived social support. Being lonely was associated with a self-reported deterioration in mental health during the pandemic (adjusted $OR$ = 3.46, 95%CI 2.03–5.91).

### Practical implications

Intervention strategies to tackle loneliness in the severe mental ill health population are needed. Further research is needed to follow-up the severe mental ill health population after pandemic restrictions are lifted to understand perceived social support and loneliness trends.

**Data Availability Statement:** We are not able to share a de-identified data set as we do not have consent from the research participants to do this.

We have checked with the GDPR team at the University of York and they have advised us that we cannot upload this data to a public repository without explicit consent from the study participants. Data requests for the full dataset may be sent to the Closing the Gap Network email: ctg-network@york.ac.uk whose steering committee manage our data requests.

**Funding:** This study is supported by the Medical Research Council, https://mrc.ukri.org/, (grant reference MR/V028529) (author SG,EP) and links with the Closing the Gap cohort, which was part-funded by the Wellcome Trust, https://wellcome.org/, (reference 204829) (author SG,EP) through the Centre for Future Health at the University of York, UK Research and Innovation, https://www.york.ac.uk/future-health/, (reference ES/S004459/1) (author SG,EP), and the NIHR Yorkshire and Humberside Applied Research Collaboration, https://www.arc-yh.nihr.ac.uk/about-us (author RW). The funders had no role in study design, data collection and analysis, decision to publish, or preparation of the manuscript.

**Competing interests:** The authors have declared that no competing interests exist.

## Originality

Loneliness was a substantial problem for the severe mental ill health population before the Covid-19 pandemic but there is limited evidence to understand perceived social support and loneliness trends during the pandemic.

## Introduction

Loneliness is increasingly recognised as a risk to mental and physical health [1], and there is evidence that levels of reported loneliness have increased during the Covid-19 pandemic [2]. Public health measures, such as physical distancing and 'shielding' (self-isolation to reduce transmission risk) has impacted the lives of the UK population. People who felt most lonely before the pandemic reported even greater loneliness after four months of lockdown [2]. However, the effects of the pandemic restrictions on the severe mental ill health (SMI) population is unknown. Loneliness is a substantial problem among people with SMI, such as bipolar or psychotic disorders, but there is limited evidence to understand the extent of loneliness and related factors in this population. Australian epidemiological studies estimate that 76–80% of people with psychosis-spectrum disorders are lonely [3, 4] which is 2.3 times higher than in the general population. However, there is no known prevalence estimates based on the UK SMI population before or during the pandemic.

Existing evidence highlights the importance of tackling loneliness in SMI. In the general population, loneliness severity is a predictor for early mortality [5, 6] and is equivalent to the health risks posed by smoking or physical inactivity [7]. In schizophrenia, loneliness is a significant contributor to lower quality of life [8, 9] and is associated with a range of negative effects, such as internalised stigma [10], lower self-esteem and self-efficacy for living in the community [11], increased symptoms of paranoia [12, 13], and increased problems such as depression [14], anxiety, and hypertension [13]. People with SMI who feel lonely are 2.69 times more likely to be admitted to inpatient psychiatric care [15].

Perceived social support (PSS) is how an individual perceives friends, family, and others as sources of material, psychological, and general support during times of need. Greater PSS is an important protective factor against loneliness. A systematic review found preliminary evidence that lower PSS is associated with worse social functioning and quality of life outcomes among people with schizophrenia and bipolar disorder [16]. In a US schizophrenia sample, greater PSS was associated with higher social functioning scores but not global functioning [16]. Lower PSS in bipolar disorder was associated with greater depression, lower functioning, and longer recovery times.

People with SMI experience additional barriers to social connectivity. Increased social stigma [17, 18], challenges presented by clinical symptoms [18], and sociodemographic factors such as greater poverty and lower likelihood of being married or in employment [19, 20] all contribute to greater loneliness among people with psychosis-spectrum disorders. It is believed that loneliness both results from, and contributes to, psychotic symptoms [21, 22]. This suggests that SMI can reduce factors such as social support which then contributes to greater loneliness. This increased loneliness can then worsen the severity of psychotic symptoms which further reduces social support [23], leading to a difficult cycle.

Being employed can be a protective factor against loneliness [24]. One study about people with schizophrenia found that being employed was associated with greater social participation but was not associated with loneliness [17]. For people with schizophrenia, reduced financial

resources could elicit feelings of shame in social encounters and not being employed can contribute to feelings of social inferiority [17]. Pre-Covid reduced employment rates among those with schizophrenia [19] could limit the protective benefits of employment on loneliness. It is not known how the pandemic may have affected employment for people with SMI.

Given the importance of loneliness as a threat to public health, and the impacts of COVID on levels of loneliness in the population, it remains important to study loneliness and associated among people with SMI. Despite the abundance of surveys exploring the psychological impacts of COVID it is a significant omission that people with SMI do not participate or are under-represented. In this study we explore the impacts of COVID restrictions on loneliness in a large clinical cohort of people with SMI.

## Methods

### Design

The Closing the Gap (CtG) study is a large (n = 9, 914) transdiagnostic clinical cohort recruited between April 2016 and March 2020. Participants have documented diagnoses of schizophrenia or delusional/psychotic illness (ICD 10 F20.X & F22.X or DSM equivalent) or bipolar disorder (ICD F31.X or DSM-equivalent). The composition of the CtG cohort has previously been described [25].

We were funded to explore the impact of the COVID-19 pandemic in a sub-section of the CtG clinical cohort and we identified participants for Optimising Well-being in Self-Isolation study (OWLS) (https://sites.google.com/york.ac.uk/owls-study/home). Recruitment and data collection to the OWLS study took place between July and December 2020. To ensure that the OWLS COVID-19 sub-cohort captured a range of demographics we created a sampling framework based on gender, age, ethnicity and whether they were recruited via primary or secondary care. OWLS participants were recruited from 17 mental health trusts and six Clinical Research Networks across urban and rural settings in England.

### Recruitment and participants

Ethical approval was granted by the Health Research Authority North West–Liverpool Central Research Ethics Committee (REC reference 20/NW/0276). To be eligible to take part in OWLS COVID-19 study, people had to be aged 18 or over, have a recorded SMI diagnosis, to have taken part in CtG study, and have consented to be contacted again to be invited to further research. This enabled us to create longitudinal data linkage and to rapidly identify participants during the COVID-19 pandemic.

### Materials

The OWLS survey took approximately 40 minutes to complete. Where possible we sought alignment of measures with a large population survey which tracked the impact of COVID on mental health [26], and with the Office of National Statistics (ONS).

**Perceived social support.**   The brief form of the Perceived Social Support Questionnaire (F-SozU K6) measures perceived social support in epidemiological contexts [27]. The six items are included in the OWLS survey and ask to what extent participants have experienced social support within the past two weeks. Scores were added to provide a total score ranging from 6 to 30, where a higher score indicates greater perceived social support.

**Loneliness.**   Loneliness was measured using the University of California, Los Angeles Loneliness Scale (UCLA-LS) 3-item [28] which asks about loneliness symptoms experienced

within the past two weeks and produces a score range of 3–9, where a higher score indicates greater loneliness.

A single item measuring loneliness was also included in the OWLS survey from the Office for National Statistics (ONS) [29] to allow comparison of findings with general population surveys. The item, "How often do you feel lonely?", had possible answers of "hardly ever"; "some of the time"; or "often".

Financial status was determined using one item in the OWLS survey, "Compared to before the pandemic restrictions, how would you say you are doing financially right now?". Responses of "I am better off" or "I am about the same" were coded as not financially worse off, and a response of "I am worse off" was coded as financially worse off. Pre-Covid-19 employment status (e.g. employed full time, student, voluntary work) was obtained from the CtG survey. Current employment status was recorded in the OWLS survey. Participants who were in full- or part-time paid employment, a student, or unpaid volunteers were coded as Professionally active and all other employment statuses were coded as Professionally inactive. Participants were considered to be shielding if they reported in the OWLS study that "I was in full isolation, not leaving my home at all" during the pandemic. Whether or not participants lived alone was determined from one item in the OWLS survey "Who lives with you?". Self-reported deterioration in mental health was determined using one OWLS survey item, "Compared with life before the beginning of the pandemic restrictions, how would you rate your [mental] health in general?". Responses of "better than before" or "about the same" were coded as no deterioration and a response of "worse than before" was coded as deterioration.

**Index of multiple deprivation.** Participant postcodes collected at the point of inception to the CtG study were used to obtain an Index of Multiple Deprivation (IMD)assigned by the Ministry of Housing, Communities and Local Government (https://imd-by-postcode. opendatacommunities.org/imd/2019). Decile scores are given between 1 and 10 and then condensed to give five possible outcomes; very high deprivation (1 and 2), high deprivation (3 and 4), moderate deprivation (5 and 6), low deprivation (7 and 8) and very low deprivation (9 and 10).

## Procedure

People who met the eligibility criteria were contacted by telephone or letter and invited to take part in the OWLS COVID-19 study. Those who agreed to take part were provided with three options: i. to carry out the survey over the phone with a researcher; ii. to be sent a link to complete the survey online; or iii. to be sent a hard copy of the questionnaire in the post to complete and return.

## Analysis

The study analysis plan was registered on Open Science Framework (available at https://osf.io/ e3kdm). The analysis plan incorrectly labelled 'perceived social support' as 'social isolation'. Analyses were undertaken using SPSS v.26. Descriptive statistics were used to describe socio-demographic characteristics, shielding status, perceived social support, and loneliness. Cronbach's alpha was used to measure internal consistency of the UCLA-LS.

To examine the associations between the independent variables (professional activity, being in shielding' status, and living alone) and perceived social support, we used multiple linear regression and we controlled for age, gender, ethnicity, socioeconomic deprivation and care setting. Although the same analysis was planned also for loneliness, the assumption of heteroscedacity was not met in the linear regression model. Therefore, we derived a binary loneliness variable, where scoring 7 and above on the UCLA-LS was considered to be lonely, and we

examined its association with the aforementioned independent variables with a binary logistic regression. Associations of all independent variables with the dependent variable were first examined with a univariable regression analysis. All independent variables were inserted all together at once in the multivariable models.

## Results

Between July and December 2020, 367 people were recruited to the OWLS study. Descriptive statistics for the sample's sociodemographic factors, shielding status, perceived social support, and loneliness are reported in Table 1. Similar rates of being lonely between the UCLA-LS (N = 125, 34.1%) and ONS (feel lonely often, N = 107, 29.2%) indicate that loneliness was measured reliably.

### Perceived social support and loneliness

There was a significant association between occupancy status and perceived social support, with those not living alone reporting greater perceived social support, adjusted B = 3.06, p < .001. Associations are presented in Table 2.

The UCLA-LS was found to be highly reliable (3 items; α = .84). Participants were more likely to report being lonely if they were living alone (adjusted OR = 2.04, 95%CI 1.212–3.431, p = .01), living in an area with high IMD (adjusted OR = 2.493, 95%CI 1.044–5.953, p = .04) and being younger in age (adjusted OR = -.98, 95%CI .964-.997, p = .02). Univariate models demonstrated that people were more likely to feel lonely if they were living in areas of very high IMD, however, this was not significantly associated in the adjusted model. Associations are presented in Table 3.

### Post-hoc analyses

A deterioration in mental health was reported by 148 (40.3%) of participants and no deterioration reported by 210 (57.2%). A logistic regression found that deterioration in mental health, after controlling for age, gender, minority-status, IMD, and care setting (primary vs secondary), was associated with being lonely (adjusted OR = 3.46, 95%CI 2.03–5.91). A multiple linear regression demonstrated that lower perceived social support, after controlling for age, gender, minority-status, IMD, and care setting (primary vs secondary), was associated with being lonely (B = -5.86, p < .001).

## Discussion

Loneliness was found to be a substantial problem for people with SMI during the pandemic; around one in three reported being lonely. This is higher than loneliness rates found in the general population during the pandemic (13–18% [30], 27% [31]). Similar patterns emerged between people with SMI and the general population; younger age and living alone were associated with greater loneliness in both populations. Lower PSS was associated with living alone. There were also similar rates of PSS between those with SMI compared to the general population (20.8 present study vs 21.6 [31]) which was associated with reduced loneliness in both studies.

The physical distancing and shielding measures introduced during the pandemic may have negatively impacted on PSS by making it more difficult to maintain social relationships. The similar patterns in loneliness between the present sample and general population could indicate that many factors which contribute to loneliness in the general population may also contribute to loneliness among those with SMI. However, given the existing literature that

**Table 1. Descriptive statistics for sociodemographic factors, shielding status, perceived social support, and loneliness.**

| Factor | N (%), total n = 367 |
|---|---|
| **Gender** | |
| Male | 187 (51) |
| Female | 174 (47.4) |
| Transgender | 6 (1.6) |
| **Age** (mean, range) | 50.5 (20–86) |
| **Ethnicity** | |
| White British | 284 (77.4) |
| Other white | 18 (4.9) |
| Mixed white / black | 5 (1.4) |
| Mixed white / Asian | 5 (1.4) |
| Other mixed | 4 (1.1) |
| Asian | 24 (6.5) |
| Black | 15 (4.1) |
| Other non-white | 12 (3.3) |
| **Index of Multiple Deprivation** | |
| Very high deprivation | 97 (26.4) |
| High deprivation | 81 (22.1) |
| Moderate deprivation | 67 (18.3) |
| Low deprivation | 55 (15) |
| Very low deprivation | 52 (14.2) |
| **Mental health care setting** | |
| Primary care | 139 (37.9) |
| Secondary care | 224 (61) |
| **Co-occupancy status** | |
| Living alone | 154 (42) |
| Not living alone | 208 (56.7) |
| **Shielding status** | |
| In full isolation, not leaving home at all | 73 (19.9) |
| Not in full isolation | 288 (78.5) |
| **F-SozU K6 Perceived social support** (mean, sd) | 20.8 (6.4) |
| **UCLA-LS Loneliness** | |
| Lonely | 125 (34.1) |
| Not lonely | 233 (63.5) |
| **ONS Loneliness** | |
| Often | 107 (29.2) |
| Some of the time | 129 (35.1) |
| Hardly ever | 122 (33.2) |
| **Professionally active before the pandemic** | |
| Yes | 123 (33.5) |
| No | 239 (65.1) |
| **Professionally active during the pandemic** | |
| Yes | 93 (25.3) |
| No | 269 (73.3) |
| **Finance during the pandemic** | |
| Being worse off | 61 (16.6) |
| Being better off | 60 (16.3) |

**Table 2. Associations between sociodemographic factors and perceived social support.**

| | Univariable model | | Multivariable model | | Multiple regression model |
|---|---|---|---|---|---|
| | B (standard error) | p | B (standard error) | p | $F(12,309) = 2.05$, $p = .02$, $R^2 = .07$ |
| **Age** | .03 (.02) | .25 | .03 (.02) | .2 | |
| **Gender** (ref: male) | | | | | |
| Female | .38 (.7) | .59 | -.06 (.72) | .93 | |
| Transgender | -4.65 (2.9) | .11 | -3.47 (2.86) | .23 | |
| **Ethnic minority** | -.25 (.96) | .79 | .35 (.99) | .72 | |
| **IMD** (ref: very low) | | | | | |
| Very high | -2 (1.11) | .08 | -.44 (1.15) | .7 | |
| High | -2.23 (1.17) | .06 | -.99 (1.19) | .41 | |
| Moderate | -2.41 (1.2) | .05 | -1.38 (1.21) | .25 | |
| Low | -.7 (1.3) | .58 | -.26 (1.24) | .84 | |
| **Currently accessing mental health services** | 1.26 (.72) | .08 | .5 (.74) | .51 | |
| **Being professionally active** | -1.04 (.8) | .195 | -.4 I.83) | .63 | |
| **Shielding** | -.27 (.82) | .74 | -.48 (.84) | .57 | |
| **Not living alone** | 3.06 (.69) | < .001 | 2.73 (.78) | < .001 | |

loneliness is a substantial problem for people with SMI, combined with the high prevalence of loneliness found in the present study, it is clear that loneliness presents a considerable problem to those with SMI. This is concerning given the strong association we found between being lonely and a deterioration in mental health.

Only a minority of participants reported a worsening to their financial wellbeing or reduction in professional activity during the pandemic. This may be because people with SMI were already disproportionately affected by socioeconomic deprivation and unemployment prior to the pandemic. Being professionally active during the pandemic was not significantly associated with PSS nor loneliness. This differs from analyses of UK-based general population studies where being economically inactive was associated with greater risk loneliness during the pandemic [32]. However, the economically-active variable from the general population analysis differed from the present study in that the general population analysis did not consider unpaid voluntary activity as being active. Further research should add context to this finding by exploring whether professional activity types (e.g. competitive paid vs voluntary activity) or settings (e.g. remote vs face-to-face working) are associated with PSS or loneliness.

Data collection occurred (Jul–Dec 2020) during continual changes to government Covid-19 policies that included both restrictions to non-essential activity and physical distancing, and also easing of restrictions and encouragement for the public to dine out in restaurants. It therefore appears most appropriate to consider the present findings in relation to an early-to-middle phase of the pandemic. Therefore, some long-term effects of the pandemic restrictions and social isolation may have influenced the present findings, but the full long-term effects were likely not felt by the time of data collection. It is important to follow-up participants into a later stage of the pandemic, and as restrictions are lifted, to explore how the long-term effects of the pandemic and restrictions affect loneliness and social isolation. It is concerning that long-term social isolation may make it more difficult to maintain relationships, thereby reducing PSS, and contribute to long-term loneliness and making it more difficult to people to return to normal post-pandemic.

Access to the internet has facilitated social communication for many during the physical distancing restrictions. However, it is not known what portion of the SMI population have access to the internet or how they interact with the internet for social communication. The

**Table 3. Associations between sociodemographic factors and loneliness.**

| | N (%) | | Univariable model | | Multivariable model | |
|---|---|---|---|---|---|---|
| | **Lonely** | **Not lonely** | **Odds ratio (95%CI)** | **p** | **Adj. Odds ratio (95%CI)** | **p** |
| **Age** | | | .99 (.97–1) | .05 | -.98 (.96–1) | .02 |
| **Gender** | | | | | | |
| Male | 68 (37.4) | 114 (62.6) | 1.32 (.85–2.05) | .22 | 1.44 (.88–2.34) | .15 |
| Female | 53 (31.2) | 117 (68.8) | 1 | | 1 | |
| **Ethnic** | | | | | | |
| **minority** | 108 (36.2) | 190 (63.8) | 1.44 (.78–2.64) | .24 | 1.97 (1.01–3.87) | .05 |
| Non-minority | 17 (28.3) | 43 (71.7) | 1 | | 1 | |
| Minority | | | | | | |
| **IMD** | | | | | | |
| Very high | 35 (36.8) | 60 (63.2) | 2.39 (1.07–5.36) | .03 | 1.52 (.64–3.6) | .35 |
| High | 34 (43) | 45 (57) | 3.1 (1.36–7.05) | .01 | 2.49 (1.04–5.95) | .04 |
| Moderate | 43 (65.2) | 23 (34.8) | 2.19 (.93–5.17) | .07 | 1.71 (.69–4.23) | .25 |
| Low | 17 (32.1) | 36 (67.9) | 1.94 (.79–4.76) | .15 | 1.84 (.73–4.62) | .2 |
| Very low | 10 (19.6) | 41 (80.4) | 1 | | 1 | |
| **Accessing** | | | | | | |
| **secondary** | 81 (36.8) | 139 (63.2) | 1.23 (.78–1.93) | .37 | 1 (.6–1.64) | .97 |
| **care** | 44 (32.1) | 93 (67.9) | 1 | | 1 | |
| Yes | | | | | | |
| No | | | | | | |
| **Professionally active** | | | | | | |
| | 28 (30.4) | 64 (69.6) | .76 (.46–1.26) | .29 | .84 (.47–1.49) | .55 |
| Yes | 97 (36.6) | 168 (63.4) | 1 | | 1 | |
| No | | | | | | |
| **Shielding** | | | | | | |
| Yes | 33 (37.9) | 54 (62.1) | 1.19 (.72–1.96) | .5 | 1.765 (1–3.11) | .5 |
| No | 92 (33.9) | 179 (66.1) | 1 | | 1 | |
| **Living alone** | | | | | | |
| Yes | 66 (43.7) | 85 (56.3) | 1.98 (1.27–3.08) | .002 | 2.04 (1.21–3.43) | .01 |
| No | 58 (28.2) | 148 (71.8) | 1 | | 1 | |

OWLS study has also explored the use of the internet and found that the majority of the present sample were limited, or non-, users of digital devices, potentially because of a lack of skills or interest [33]. This limited internet access could therefore be contributing to greater loneliness experienced during the pandemic. Further study should explore whether digital interventions in this population are a viable means of improving PSS and reducing loneliness among those with SMI, particularly when face-to-face communication is limited.

## Limitations

It would have been preferable to have a pre-Covid profile of the measured variables, but this was a cross-sectional study so there was no pre-Covid baseline measure. It was therefore not possible to understand changes to loneliness during the pandemic. We plan to track trends in the measured variables over time to see the longitudinal course.

The shielding variable did not account for individuals who were shielding and living alone, compared to those who were shielding and not living alone. This may account for the lack of association between shielding and PSS or loneliness.

Loneliness and PSS may have varied between participants with bipolar disorder and psychosis spectrum disorder due to the effects of the disorders. However, this study only explored SMI as a group and future study should explore potential variations in findings between diagnoses. Different coping strategies between diagnosis groups should also be explored, for example, how the internet may be used for social communication.

## Conclusion

The Covid-19 public health measures have increased barriers to social connectivity that has increased loneliness among the general public. Pre-existing barriers to social connectivity for people with SMI meant that loneliness was already a substantial problem. Once the pandemic restrictions are removed and barriers to socialising are reduced for the general population then the pre-existing barriers unique to people with SMI will likely remain. There is a risk that loneliness rates may remain higher among those with SMI than the general population and this will exacerbate health inequalities. Further research should follow-up people with SMI as the pandemic restrictions are lifted to understand loneliness trends. Additional study is also needed to understand the barriers to social connectivity for people with SMI, and to understand the best strategies to intervene. Specifically, the internet and digital connectivity should be explored as potential strategies to tackle problems of PSS and loneliness. Theoretical models of loneliness that apply to the general population likely also apply to those with SMI [18], so research should explore the effectiveness of general strategies to reduce loneliness for people with SMI. Intervention strategies may be adapted to tackle the unique barriers experienced by those with SMI. An intervention that is tailored to young adults who live alone may be an effective response to address the main burden of loneliness among people with SMI. Further understanding of loneliness and its relation to mental health among people with SMI is needed to develop this area of research.

## Acknowledgments

We thank the participants in the OWLS study and NHS mental health staff for their support with this study. We would like to thank the lived experience advisors who contributed their time and lived expertise to this study.

## Author Contributions

**Conceptualization:** Paul Heron, Panagiotis Spanakis, Suzanne Crosland, Gordon Johnston, Elizabeth Newbronner, Ruth Wadman, Lauren Walker, Simon Gilbody, Emily Peckham.

**Formal analysis:** Panagiotis Spanakis, Emily Peckham.

**Funding acquisition:** Paul Heron, Panagiotis Spanakis, Elizabeth Newbronner, Simon Gilbody, Emily Peckham.

**Investigation:** Paul Heron, Panagiotis Spanakis, Suzanne Crosland, Lauren Walker, Emily Peckham.

**Methodology:** Paul Heron, Panagiotis Spanakis, Suzanne Crosland, Gordon Johnston, Elizabeth Newbronner, Ruth Wadman, Lauren Walker, Simon Gilbody, Emily Peckham.

**Project administration:** Paul Heron, Panagiotis Spanakis, Emily Peckham.

**Supervision:** Simon Gilbody, Emily Peckham.

**Writing – original draft:** Paul Heron.

**Writing – review & editing:** Paul Heron, Panagiotis Spanakis, Suzanne Crosland, Gordon Johnston, Elizabeth Newbronner, Ruth Wadman, Lauren Walker, Simon Gilbody, Emily Peckham.

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
