## [Decision Letter · Decision Letter 0]

20 Sep 2021

PONE-D-21-22036Loneliness among people with severe mental ill health during the COVID-19 pandemic: results from a linked UK population cohort studyPLOS ONE

Dear Dr. Heron,

Thank you for submitting your manuscript to PLOS ONE. After careful consideration, we feel that it has merit but does not fully meet PLOS ONE’s publication criteria as it currently stands. Therefore, we invite you to submit a revised version of the manuscript that addresses the points raised during the review process.

ACADEMIC EDITOR: Considering the reviewers comments and my own reading of the paper, I am suggesting a major revision for this paper. Third reviewer who has not submitted full review but saved partial review with following comment. 

"The study is timely but lacks methodological and statistical rigor", try to respond to this comment as well. 

We look forward to receiving your revised manuscript.

Kind regards,

Srinivas Goli, Ph.D.

Academic Editor

PLOS ONE

Journal Requirements:

Additional Editor Comments (if provided):

Considering the reviewers comments and my own reading of the paper, I am suggesting a major revision for this paper. Third reviewer who has not submitted full review but saved partial review with following comment.

"The study is timely but lacks methodological and statistical rigor", try to respond to this comment as well.

Reviewers' comments:

Reviewer's Responses to Questions

**Comments to the Author**

1. Is the manuscript technically sound, and do the data support the conclusions?

Reviewer #1: Yes

Reviewer #2: Yes

2. Has the statistical analysis been performed appropriately and rigorously? 

Reviewer #1: Yes

Reviewer #2: Yes

3. Have the authors made all data underlying the findings in their manuscript fully available?

Reviewer #1: Yes

Reviewer #2: No

4. Is the manuscript presented in an intelligible fashion and written in standard English?

Reviewer #1: Yes

Reviewer #2: Yes

5. Review Comments to the Author

Reviewer #1: The topic of research is very interesting and it needs to be published. Nevertheless, I would like to raise the following issues which I considered is good to be taken for the betterment of the paper.

GENERAL

Title: I personally recommend you to take minor revision on your title of the research because it is more comfortable for readers if it can be revised as ‘Loneliness among people with severe mental illness during the COVID-19 pandemic: results from a linked UK population cohort study”

Abstract

Purpose: The title of your research needs to have complete information about what you are going to do. Your title is not sufficient to describe what is mentioned in your purpose of study section. It needs revision; put your exact objective with respect to your topic of interest.

Introduction: interesting

Methods: interesting

Result: Table 2 is out of what you are intended to do; beyond your objective

Discussion: very interesting but too little when compared to what you have reported in the result section and inadequate scientific evidences. Needs revision

References: there are a number of web link citations in the main document; it is good if you use a uniform referencing style.

Reviewer #2: I am very grateful to the Editors that I have the opportunity to revise this manuscript.

Below is my review report:

-The statistic analyzes are well done.

Formal problem:

-IMDD: the meaning of the abbreviation is not explained in the text. If it is Index of Multiple Deprivation, than in Table 1 there is „moderate deprivation” and inTable 2,3 the same category called „medium”. It will be better to check nomenclature and use one name.

-There is a reference form error on page 10: Mishu 2019- I can not find in reference list, and the form is problematic in the text.

Questions or suggestions about method:

-When did you do the reports? I think it can be important, that study sampling was in first “COVID pandemic wave” (2020 spring) or after half/ one year from its beginning. I think mental status could be deterioration after several months of lockdown.

It is interesting finding, regarding the timing of research: “Only a minority of participants reported a worsening to their financial wellbeing or reduction in professional activity during the pandemic.This may be because people with SMI were already disproportionately affected by socioeconomic deprivation and unemployment prior to the pandemic.” Now, among the waves of the COVID pandemic of the past 1,5 years, is this research result still the same or only at the beginning of the pandemic was?

-In this study severe mental ill health is treated as a homogeneous group, it is a question for me that patients with different type of mental illnesses (and different coping ability or pss) can be treat like a homogeneous group.

-I would have also asked in questionare about (online) varieties of social support because just as the general population has replaced personal relationships with this, I think study group as well.

Discussion:

As I read the results and discussion, I felt that the study has the result what I expected at beginning of the introduction part. With the research results, the authors fulfilled the described study goal regarding the explore perceived social support and loneliness and factor associations during the Covid-19 pandemic, but there was no surprising finding at the end. I would have liked to read about what coping methods were, and in more detail what deterioration of loneliness was associated with their patients mental health in lockdown period. Maybe the study questionaire was not enough detailed or targetted, and that is why there is not a breakthrough result.

Conclusion:

“Additional study is also needed to understand the barriers to social connectivity for people with SMI, and to understand the best strategies to intervene.” Correct, and important conclusion. I hope you will continue your study and we will read about the barriers to social connectivity and your strategies to intervene.

6. PLOS authors have the option to publish the peer review history of their article (what does this mean?). If published, this will include your full peer review and any attached files.

Reviewer #1: **Yes: **MENGESHA SRAHBZU BIRESAW

Reviewer #2: No

---

## [Author Response · Author response to Decision Letter 0]

26 Oct 2021

Thank you for reviewing the manuscript and for providing us with feedback to improve the manuscript. We have responded to each of the reviewer comments in the attached document 'Response to Reviewers' alongside the cover letter. I have also pasted the text below.

Dear Plos Editorial team and reviewers,

Thank you for processing the submitted manuscript and for providing us with the opportunity to revise the submitted manuscript. We thank the reviewers for their thoughtful comments which have helped us to revise and improve the manuscript. 

We have addressed each of the reviewers’ comments and expanded the manuscript’s discussion section. Please find our detailed responses to each of the comments below in blue text. We hope that the manuscript may now be suitable for publication, but please do not hesitate to contact us if there are matters to address, or if we can provide additional information. 

Kind wishes,

Paul Heron

Comments to the Author

Considering the reviewers comments and my own reading of the paper, I am suggesting a major revision for this paper. Third reviewer who has not submitted full review but saved partial review with following comment.

"The study is timely but lacks methodological and statistical rigor", try to respond to this comment as well.

Author response: We have addressed a problem with the study data being publicly available and a problem with the analysis plan in the comments below. The discussion section has been expanded and adds context to the timing of data collection. These points are described in further detail below in response to reviewer comments. 

1. Is the manuscript technically sound, and do the data support the conclusions?

Reviewer #1: Yes

Reviewer #2: Yes

Author response: Thank you

2. Has the statistical analysis been performed appropriately and rigorously?

Reviewer #1: Yes

Reviewer #2: Yes

Author response: Thank you

3. Have the authors made all data underlying the findings in their manuscript fully available?

Reviewer #1: Yes

Reviewer #2: No

Author response: We are not able to share a de-identified data set as we do not have consent from the research participants to do this. We have checked with the GDPR team at the University of York and they have advised us that we cannot upload this data to a public repository without explicit consent from the study participants. Data requests for the full dataset may be sent to the Closing the Gap Network email: ctg-network@york.ac.uk whose steering committee manage our data requests. 

4. Is the manuscript presented in an intelligible fashion and written in standard English?

Reviewer #1: Yes

Reviewer #2: Yes

Author response: Thank you

5. Review Comments to the Author

Reviewer #1: The topic of research is very interesting and it needs to be published. Nevertheless, I would like to raise the following issues which I considered is good to be taken for the betterment of the paper.

GENERAL

Title: I personally recommend you to take minor revision on your title of the research because it is more comfortable for readers if it can be revised as ‘Loneliness among people with severe mental illness during the COVID-19 pandemic: results from a linked UK population cohort study”

Author response: We have updated the title as recommended

Abstract

Purpose: The title of your research needs to have complete information about what you are going to do. Your title is not sufficient to describe what is mentioned in your purpose of study section. It needs revision; put your exact objective with respect to your topic of interest.

Author response: We have updated the ‘Purpose’ section of the abstract to provide a more complete picture of the manuscript. Taken with the above comment that suggests a minor amendment to the title, we believe that this comment refers to the ‘Purpose’ section of the abstract rather than the manuscript title. We apologise if we have misinterpreted this comment and would be happy to amend the manuscript title further. 

Introduction: interesting

Methods: interesting

Result: Table 2 is out of what you are intended to do; beyond your objective

Author response: Thank you for highlighting this. The analysis plan contains an error where it refers to the variable as ‘Social isolation’ rather than ‘Perceived social support’. This can be corroborated by the survey being uploaded prior to analysis which contains a measure of Perceived social support, but no measure of Social isolation (link to survey in analysis plan: https://osf.io/8qwhd/#!). We have added information to the ‘Analysis’ section of the manuscript to clarify this. 

Discussion: very interesting but too little when compared to what you have reported in the result section and inadequate scientific evidences. Needs revision

Author response: We have expanded the discussion section. In particular, we discuss two points, mentioned in the below comments; the internet and digital connectivity for social contact, and adding context to when the data was collected in relation to pandemic restrictions. 

References: there are a number of web link citations in the main document; it is good if you use a uniform referencing style.

Author response: We have updated the in-text weblinks to maintain consistency

Reviewer #2: I am very grateful to the Editors that I have the opportunity to revise this manuscript.

Below is my review report:

-The statistic analyzes are well done.

Formal problem:

-IMDD: the meaning of the abbreviation is not explained in the text. If it is Index of Multiple Deprivation, than in Table 1 there is „moderate deprivation” and inTable 2,3 the same category called „medium”. It will be better to check nomenclature and use one name.

Author response: We have updated the text to explain the abbreviation, and to refer to it as Index of Multiple Deprivation (IMD), rather than Index of Multiple Deprivation Decile (IMDD) which was incorrect. The text also now consistently refers to the middle category as ‘moderate’ deprivation.

-There is a reference form error on page 10: Mishu 2019- I can not find in reference list, and the form is problematic in the text.

Author response: We have updated the reference list and article body to add the Mishu 2019 reference. Other references have also been re-numbered to allow this. 

Questions or suggestions about method:

-When did you do the reports? I think it can be important, that study sampling was in first “COVID pandemic wave” (2020 spring) or after half/ one year from its beginning. I think mental status could be deterioration after several months of lockdown.

It is interesting finding, regarding the timing of research: “Only a minority of participants reported a worsening to their financial wellbeing or reduction in professional activity during the pandemic.This may be because people with SMI were already disproportionately affected by socioeconomic deprivation and unemployment prior to the pandemic.” Now, among the waves of the COVID pandemic of the past 1,5 years, is this research result still the same or only at the beginning of the pandemic was?

Author response: Thank you for this. We have added detail to the discussion to add context to these findings and discuss how long-term effects of restrictions may have influenced the findings. 

-In this study severe mental ill health is treated as a homogeneous group, it is a question for me that patients with different type of mental illnesses (and different coping ability or pss) can be treat like a homogeneous group.

Author response: We were also interested to explore differences between diagnoses but did not want to conduct too many post-hoc analyses. We have now added detail to the ‘Limitations’ section to discuss this. 

-I would have also asked in questionare about (online) varieties of social support because just as the general population has replaced personal relationships with this, I think study group as well.

Author response: Thank you, we have now added detail to the discussion about the role of internet and digital devices in social communication during the pandemic among people with SMI. Digital connectivity and internet use was explored as part of the OWLS study in a separate article, so we have referred readers to these articles for further information. 

Discussion:

As I read the results and discussion, I felt that the study has the result what I expected at beginning of the introduction part. With the research results, the authors fulfilled the described study goal regarding the explore perceived social support and loneliness and factor associations during the Covid-19 pandemic, but there was no surprising finding at the end. I would have liked to read about what coping methods were, and in more detail what deterioration of loneliness was associated with their patients mental health in lockdown period. Maybe the study questionaire was not enough detailed or targetted, and that is why there is not a breakthrough result.

Author response: Thank you. Due to the continually changing nature of the pandemic, and how concerning we found these findings to be, we felt it was important to publish these results early. We hope they will generate interest and discussion around the topic. In time, we plan to publish the OWLS study’s follow-up data to explore long-term trends in loneliness and social support, and to explore these topics with qualitative methods. 

Conclusion:

“Additional study is also needed to understand the barriers to social connectivity for people with SMI, and to understand the best strategies to intervene.” Correct, and important conclusion. I hope you will continue your study and we will read about the barriers to social connectivity and your strategies to intervene.

Author response: Thank you.

---

## [Decision Letter · Decision Letter 1]

16 Nov 2021

PONE-D-21-22036R1Loneliness among people with severe mental illness during the COVID-19 pandemic: results from a linked UK population cohort studyPLOS ONE

Dear Dr. Heron,

Thank you for submitting your manuscript to PLOS ONE. After careful consideration, we feel that it has merit but does not fully meet PLOS ONE’s publication criteria as it currently stands. Therefore, we invite you to submit a revised version of the manuscript that addresses the points raised during the review process.

ACADEMIC EDITOR: Considering the reviewers suggestion, I am going with a decision of minor revision. 

We look forward to receiving your revised manuscript.

Kind regards,

Srinivas Goli, Ph.D.

Academic Editor

PLOS ONE

Journal Requirements:

Additional Editor Comments:

Considering the reviewers suggestion, I am going with a decision of minor revision.

Reviewers' comments:

Reviewer's Responses to Questions

**Comments to the Author**

1. If the authors have adequately addressed your comments raised in a previous round of review and you feel that this manuscript is now acceptable for publication, you may indicate that here to bypass the “Comments to the Author” section, enter your conflict of interest statement in the “Confidential to Editor” section, and submit your "Accept" recommendation.

Reviewer #1: All comments have been addressed

Reviewer #2: All comments have been addressed

2. Is the manuscript technically sound, and do the data support the conclusions?

Reviewer #1: Yes

Reviewer #2: Yes

3. Has the statistical analysis been performed appropriately and rigorously? 

Reviewer #1: Yes

Reviewer #2: Yes

4. Have the authors made all data underlying the findings in their manuscript fully available?

Reviewer #1: Yes

Reviewer #2: Yes

5. Is the manuscript presented in an intelligible fashion and written in standard English?

Reviewer #1: Yes

Reviewer #2: Yes

6. Review Comments to the Author

Reviewer #1: THANK YOU very much for addressing my concerns for the betterment of the paper. in its current status it has became an interesting paper.

Reviewer #2: Dear Author!

Thank you for your revised manuscript and answers for my questions.

Your manuscript is still timely, but need minor revision.

My review report:

Technical problem:

Reference 33: There are 9 authors mentioned, and as I know you should write max. 6 then abbreviation et al.- Please look after it, and correct.

1. Abstract: Purpose: Aim/ goal/ purpose - the term- should be use in drafting this part of manuscript - please revise (content is ok)

2. Introduction, result part do not need further revision- in my opinion.

3. Methods: Data collection timing should be here mentioned firstly.

4. Discussion

Discussion needs revision. The main message of your study and the results should be in focus, and should be detailed here. Now the new paragraphs seem to be too detailed- as I see the whole discussion, and the results. Conditions of data collection and its timing need to be shorten here. Use of internet by SMI patient is very interesting, and it is good to read about it, but not involved in this study- so it is an additional information, need compress too.

7. PLOS authors have the option to publish the peer review history of their article (what does this mean?). If published, this will include your full peer review and any attached files.

Reviewer #1: **Yes: **MENGESHA SRAHBZU BIRESAW

Reviewer #2: No

---

## [Author Response · Author response to Decision Letter 1]

20 Dec 2021

Dear PLOS ONE Editorial team and Reviewers,

On behalf of the authors, thank you to the editors and reviewers. We appreciate your feedback to refine and improve the article. We have updated the article in line with the recommendations. Please see the relevant 'Author response' to each point below. 

We hope that the article may be ready for acceptance. Please do not hesitate to contact me if you would like to discuss the manuscript further. 

Kind wishes,

Paul

Feedback from PLOS ONE

Journal Requirements:

Author response: The reference list has been updated to correct errors. 

Additional Editor Comments:

Considering the reviewers suggestion, I am going with a decision of minor revision.

Reviewers' comments:

Reviewer's Responses to Questions

Comments to the Author

1. If the authors have adequately addressed your comments raised in a previous round of review and you feel that this manuscript is now acceptable for publication, you may indicate that here to bypass the “Comments to the Author” section, enter your conflict of interest statement in the “Confidential to Editor” section, and submit your "Accept" recommendation.

Reviewer #1: All comments have been addressed

Reviewer #2: All comments have been addressed

2. Is the manuscript technically sound, and do the data support the conclusions?

Reviewer #1: Yes

Reviewer #2: Yes

3. Has the statistical analysis been performed appropriately and rigorously?

Reviewer #1: Yes

Reviewer #2: Yes

4. Have the authors made all data underlying the findings in their manuscript fully available?

Reviewer #1: Yes

Reviewer #2: Yes

5. Is the manuscript presented in an intelligible fashion and written in standard English?

Reviewer #1: Yes

Reviewer #2: Yes

6. Review Comments to the Author

Reviewer #1: THANK YOU very much for addressing my concerns for the betterment of the paper. in its current status it has became an interesting paper.

Reviewer #2: Dear Author!

Thank you for your revised manuscript and answers for my questions.

Your manuscript is still timely, but need minor revision.

My review report:

Technical problem:

Reference 33: There are 9 authors mentioned, and as I know you should write max. 6 then abbreviation et al.- Please look after it, and correct.

Author response: This has been corrected.

1. Abstract: Purpose: Aim/ goal/ purpose - the term- should be use in drafting this part of manuscript - please revise (content is ok)

Author response: The terms in this section have been updated.

2. Introduction, result part do not need further revision- in my opinion.

Author response: Thank you.

3. Methods: Data collection timing should be here mentioned firstly.

Author response: The methods have now been updated with the recruitment and data collection timing.

4. Discussion

Discussion needs revision. The main message of your study and the results should be in focus, and should be detailed here. Now the new paragraphs seem to be too detailed- as I see the whole discussion, and the results. Conditions of data collection and its timing need to be shorten here. Use of internet by SMI patient is very interesting, and it is good to read about it, but not involved in this study- so it is an additional information, need compress too.

Author response: The discussion section has had some detail removed to make it more succinct. This will also mean that the focus of the discussion is placed more on the main findings.

---

## [Editor Report · Decision Letter 2]

23 Dec 2021

Loneliness among people with severe mental illness during the COVID-19 pandemic: results from a linked UK population cohort study

PONE-D-21-22036R2

Dear Dr. Heron,

We’re pleased to inform you that your manuscript has been judged scientifically suitable for publication and will be formally accepted for publication once it meets all outstanding technical requirements.

Kind regards,

Srinivas Goli, Ph.D.

Academic Editor

PLOS ONE

Additional Editor Comments (optional):

Revisions are satisfactory, thus I am recommending this article for publication in PLOS One. However, before publishing please ask authors to format the Tables and the paper in line with PLOS One guidelines. They are not looking good in its present form.
---

## [Editor Report · Acceptance letter]

4 Jan 2022

PONE-D-21-22036R2 

Loneliness among people with severe mental illness during the COVID-19 pandemic: results from a linked UK population cohort study 

Dear Dr. Heron:

I'm pleased to inform you that your manuscript has been deemed suitable for publication in PLOS ONE. Congratulations! Your manuscript is now with our production department. 

Kind regards, 

on behalf of

Dr. Srinivas Goli 

Academic Editor

PLOS ONE